# Engineering of the endogenous *HBD* promoter increases HbA2

Mandy Y Boontanrart*, Elia Mächler, Simone Ponta, Jan C Nelis, Viviana G Preiano, Jacob E Corn*

Department of Biology, ETH Zurich, Zurich, Switzerland

**Abstract** The β-hemoglobinopathies, such as sickle cell disease and β-thalassemia, are one of the most common genetic diseases worldwide and are caused by mutations affecting the structure or production of β-globin subunits in adult hemoglobin. Many gene editing efforts to treat the β-hemoglobinopathies attempt to correct β-globin mutations or increase γ-globin for fetal hemoglobin production. δ-globin, the subunit of adult hemoglobin A2, has high homology to β-globin and is already pan-cellularly expressed at low levels in adult red blood cells. However, upregulation of δ-globin is a relatively unexplored avenue to increase the amount of functional hemoglobin. Here, we use CRISPR-Cas9 to repair non-functional transcriptional elements in the endogenous promoter region of δ-globin to increase overall expression of adult hemoglobin 2 (HbA2). We find that insertion of a KLF1 site alone is insufficient to upregulate δ-globin. Instead, multiple transcription factor elements are necessary for robust upregulation of δ-globin from the endogenous locus. Promoter edited HUDEP-2 immortalized erythroid progenitor cells exhibit striking increases of *HBD* transcript, from less than 5% to over 20% of total β-like globins in clonal populations. Edited CD34 +hematopoietic stem and progenitors (HSPCs) differentiated to primary human erythroblasts express up to 46% *HBD* in clonal populations. These findings add mechanistic insight to globin gene regulation and offer a new therapeutic avenue to treat β-hemoglobinopathies.

## Editor's evaluation

This study presents the important finding that gene editing could be used to activate δ-globin expression to treat disorders of red blood cell synthesis. The evidence supporting the claims of the authors is convincing, particularly in the clonal cell lines. The data show this approach to have promise and identify avenues of effort that could be pursued to advance it to a clinical strategy for hemoglobinopathy treatment.

*For correspondence:
mandy.boontanrart@biol.ethz.ch (MYB);
jacob.corn@biol.ethz.ch (JEC)

**Competing interest:** The authors declare that no competing interests exist.

## Introduction

Red blood cells, also known as erythrocytes, are packed with hemoglobin and circulate the body to supply all organs with oxygen. Hemoglobin is a hetero-tetrameric protein made up of two α-like and two β-like subunits. Hemoglobin A1 (HbA1) accounts for approximately 97% of the hemoglobin expressed in adults and is composed of two α-globin subunits (*HBA*) and two β−globin (*HBB*) subunits. Hemoglobin A2 (HbA2) accounts for the remaining 2–3% of hemoglobin expressed in adults and is composed of two α-globin subunits and two δ-globin (*HBD*) subunits *Steinberg and Rodgers, 2015*.

   The β-hemoglobinopathies, such as sickle cell disease (SCD) and β-thalassemia, are caused by mutations in *HBB* that effect the structure or expression of β-globin. The clinical hallmarks include hemolytic anemia and vaso-occlusion, which can lead to acute and chronic pain and organ damage. Clinical management is limited to frequent blood transfusions and life-long treatment of anemia and pain crises. The only curative approach is allogeneic stem cell transplantation, which is dependent

upon HLA-identical donor availability *Locatelli et al., 2013*. Fetal hemoglobin (HbF), which is the predominant hemoglobin expressed before birth, has anti-sickling properties and its re-expression is frequently pursued as a treatment for β-hemoglobinopathies *Wienert et al., 2018*. While increasing HbF has shown to be clinically effective to combat SCD, studies have also validated the in vitro and in vivo anti-sickling abilities of δ-globin using a humanized mouse model of SCD *Nagel et al., 1979*; *Poillon et al., 1993*; *Porcu et al., 2021*; *Waterman et al., 1979*.

Increased HbA2 expression has some potential advantages over HbF that suggests it could provide an alternative avenue for ameliorating the β-hemoglobinopathies. HbA2 is weakly transcriptionally active and expressed pancellularly in all adult red blood cells *Heller and Yakulis, 1969*; *Steinberg, 2021*. Additionally, δ-globin shares 93% amino acid homology to β-globin, suggesting that HbA2 may be a better replacement relative to HbF, whose γ-globin subunit shares 73% amino acid homology to β-globin. Finally, HbF is known to bind oxygen more tightly than HbA1, an evolutionary advantage for the fetus to draw oxygen from the maternal blood source, while HbA2 has a similar oxygen-binding capacity as HbA1 *Di Cera et al., 1989*; *Inagaki et al., 2000*. On-going clinical trials re-express HbF to extremely high levels not commonly seen even in individuals with Hereditary Persistence of Fetal Hemoglobin *Frangoul et al., 2021*. The effect of extreme maternal HbF re-expression during pregnancy is currently unknown.

The genes for β−globin (*HBB*) and δ-globin (*HBD*) are located in the β-like globin cluster and regulated by the same control region. The β-like globin cluster is located on chromosome 11, and harbors the five β-like genes: *HBB* (β-globin gene), *HBD* (δ-globin gene), *HBG1* and *HBG2* (γ-globin genes), and HBE (ε-globin gene). β-globin and δ-globin both comprise 147 amino acids and differ at only 10 positions *Moleirinho et al., 2013*. The extreme difference in expression levels between these two globins is not due to protein instability or differences in translation, but instead results from a lower transcription rate *Steinberg and Rodgers, 2015*. The globin genes are arranged in order of their expression during development and regulated by contact to a distal enhancer called the Locus Control Region that contains five active DNase Hypersensitivity Sites *Li et al., 2002*. A comparative genomics study has shown that, compared to the *HBB* promoter, the *HBD* promoter has mutations in multiple transcriptional elements including a KLF1, NF-Y, β-DRF, and TFIIB binding site (*Figure 1A*) *Zaldívar-López et al., 2017*.

Previous studies using transgenic approaches have shown that inclusion of a KLF1 motif in the *HBD* promoter can drive exogenous expression of δ-globin *Porcu et al., 2021*; *Donze et al., 1996*; *Ristaldi et al., 1999*. However, these studies do not reflect the complex chromosomal context and extensive epigenetic regulation at the β-like globin cluster. Due to the large size of the β-like globin locus, transgenic studies have mostly included only a subset of the genes of the β-like globin locus and a minimal region of the LCR. They therefore do not necessarily predict the biological outcomes of perturbations at the native β-globin locus *Woodard et al., 2022*.

The expression of globin genes is a tightly regulated developmental process. There are currently no drugs or therapeutic approaches to increase HbA2 for the β-hemoglobinopathies. Using CRISPR-Cas9 genome editing, we used homology directed repair at the endogenous *HBD* promoter to engineer the transcriptional elements present in *HBB*. We find that insertion of single transcriptional elements to the endogenous promoter is insufficient for δ-globin upregulation. However, insertion of KLF1, β-DRF, and TFIIB motifs drive high expression of δ-globin from the endogenous locus in clonal populations of HUDEP-2 cells and primary erythroblasts. This leads to reconstitution of high levels of HbA2, over 10-fold increase compared to WT unedited cells. Our work adds mechanistic insight to the globin gene regulation at the β-like globin cluster and suggests a potential therapeutic avenue to upregulate HbA2 for the β-hemoglobinopathies.

## Results

### Targeting the endogenous *HBD* promoter to introduce functional *HBB* promoter elements

We aligned the promoter sequences of *HBB* and *HBD* to identify transcriptional elements missing in the *HBD* promoter (*Figure 1a*). This highlighted multiple mutations and deletions in the KLF1, NF-Y, β-DRF, and TFIIB binding motifs. To re-engineer the endogenous *HBD* promoter, we employed CRISPR-Cas9 induced homology directed repair (HDR) gene editing *Jinek et al., 2012*; *Yeh et al., 2019*.

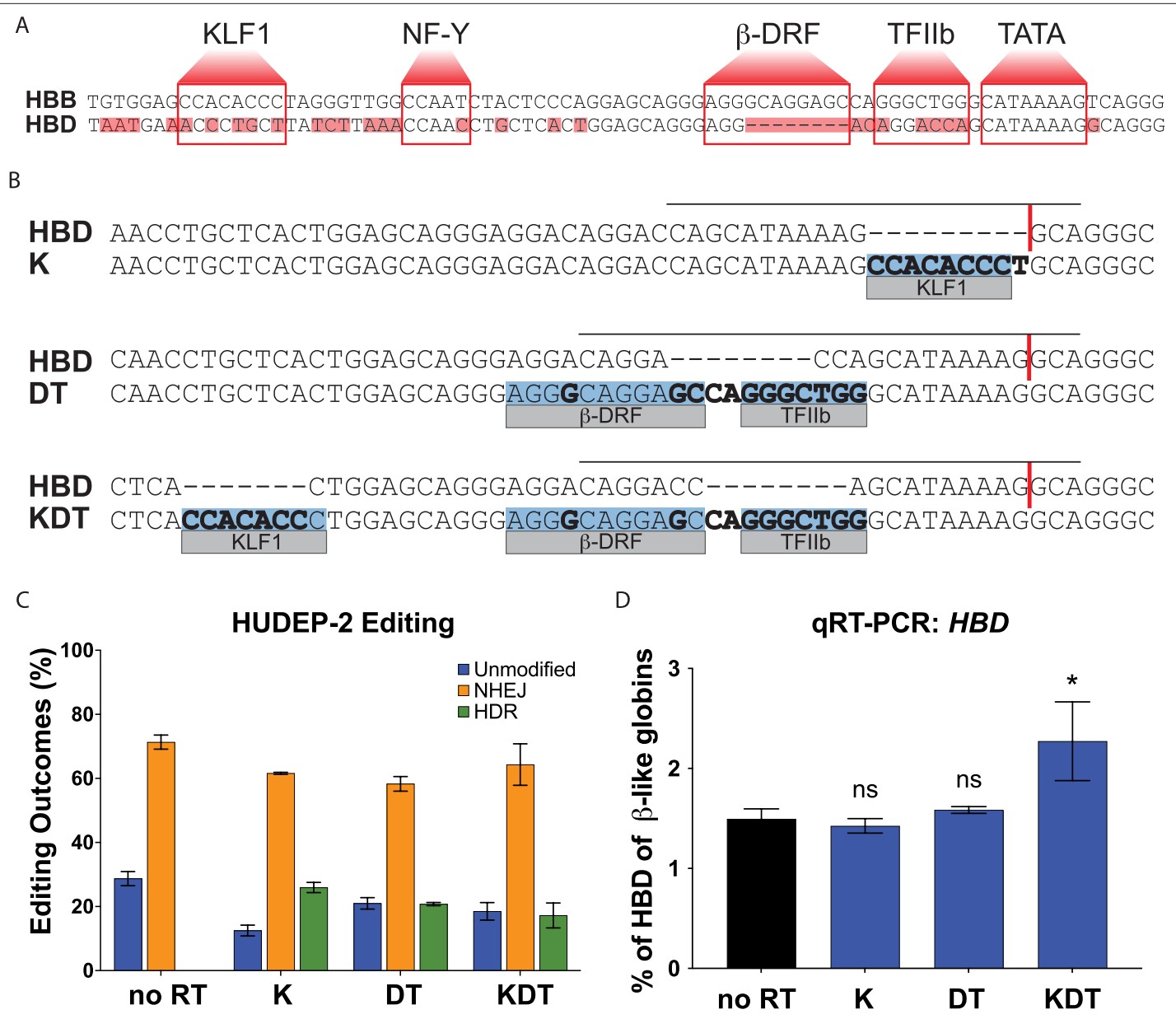

**Figure 1.** Targeting and design of the endogenous *HBD* promoter. (**A**) Alignment of the *HBB* and *HBD* promoter sequences. Transcription factor binding sequences for KLF1, NF-Y, β-DRF, TFIIb, and TATA are shown in boxes and base pair mismatches between *HBB* and *HBD* are highlighted in red. (**B**) The repair template designs for insertions of KLF1 (**K**), β-DRF and TFIIb (**DT**), and KLF1, β-DRF, and TFIIb (**KDT**) directly compared to the *HBD* promoter. The inserted transcription factor binding sequences are highlighted in blue. Any base pair changes are in bold. The gRNA is indicated by a black horizontal line and the cut site is indicated by a red vertical line. (**C**) HUDEP-2 editing efficiencies showing percentages of unmodified, NHEJ, or HDR alleles. Conditions tested were Cas9 and sgRNA RNP with no repair template (no RT), K, DT, and KDT repair templates. This experiment was performed three times and the data is presented as mean ± SD of three biological replicates. (**D**) qRT-PCR of *HBD* after pooled editing of HUDEP-2 cells with no RT, K, DT, and KDT and 5 days of differentiation. Data is plotted as % of all β-like globins (*HBB*, *HBG1/2*, *HBD*). The three biological replicates from the editing experiment in (**C**) were each differentiated and the data is presented as mean ± SD of three biological replicates. p Value indicates paired, two-tailed student *t* test (ns, non-significant; *, p≤0.05; **, p≤0.01).

The online version of this article includes the following figure supplement(s) for figure 1:

**Figure supplement 1.** Genotypes of HUDEP-2 clones with heterozygous or homozygous knock-ins of KLF1, β-DRF, and TFIIb sequences.

We designed and tested three sgRNAs targeting the *HBD* promoter (*Supplementary file 1*) and three HDR templates that would incorporate base pairs needed to complete the transcriptional element motifs (*Figure 1b*). The HDR templates were designed as single stranded oligo donor nucleotides (ssODN) to either insert a KLF1 (K) sequence, a β-DRF and TFIIB (DT) sequence, or all three elements (KDT). A previous transgenic study showed that the NF-Y site has a lesser impact on *HBB* transcription in comparison to KLF1 *Tang et al., 1997*. Therefore, in order to minimize the number of transcription element iterations for testing, we omitted NF-Y from our designs. For the K repair template, the KLF1 sequence was inserted adjacent to the cut site, after the TATA box sequence, in order to disrupt the seed sequence of the sgRNA to reduce subsequent cutting post HDR repair. For the DT repair template, β-DRF and TFIIB motifs were designed into a single template as they are separated by only 2 base pairs. For the KDT repair template, we maintained the ordering between motifs KLF1, β-DRF and TFIIB to mimic the *HBB* promoter and limited the number of mutated base pairs to increase the likelihood of successful HDR editing. For both the K and KDT repair template, the KLF1 motifs inserted are not designed in the homologous position to the KLF1 site in the *HBB* promoter. The homologous site is considerably upstream of the sgRNA cut site and therefore more difficult to integrate given the relatively short conversion tracts of HDR in human cells.

We performed editing in HUDEP-2 cells, an immortalized cell line capable of differentiating into hemoglobin-producing erythroid cells *Kurita et al., 2013*, using Cas9 ribonucleoprotein (RNP) and an ssODN. After optimizing conditions and testing multiple sgRNAs (*Supplementary file 1*; data not shown), we found that the sgRNA g1 had the highest editing efficiency, and proceeded to use sgRNA g1 for all further experiments. Editing resulted in NHEJ rates ranging from 56% to 74% and HDR efficiencies from 13% to 27% as measured by next generation sequencing (NGS; *Figure 1c*). The edited pools were differentiated to erythroblasts and qRT-PCR measurements were taken to assess the effects on *HBD* expression (*Figure 1d*). Despite roughly equal HDR rates between all promoter designs, we observed a statistically significant increase in *HBD* only for the KDT design with three elements.

## Homozygous knock-in of KLF1, TFIIB, and β-DRF leads to robust increase of *HBD* in HUDEP-2 cells

Heterogeneously edited pools of cells, containing a mixture of alleles, can mask large effects at a clonal level. We isolated heterozygous and homozygous HDR clones to more accurately assess the effect of each motif edit on *HBD* expression. We obtained at least three heterozygotes and homozygotes for each knock-in construct, as verified by amplicon NGS sequencing of the *HBD* promoter (*Figure 1—figure supplement 1*). We used ChIP-qPCR to test whether the inserted motifs were functional in recruiting KLF1 and RNA Pol II to the *HBD* promoter, comparing to the *HBB* promoter and *VEGFA* as a positive and negative control, respectively. Due to the unavailability of a TFIIB antibody suitable for ChIP, we performed ChIP for RNA Pol II (*Figure 2a*). While Pol II is a binding partner of TFIIB in the pre-initiation complex, this ChIP does have the limitation of indirectly immunoprecipitating other factors in the transcriptional pre-initiation complex and might also reflect general promoter activity. The β-DRF motif has been shown to be important for high transcription of *HBB*, but its bona fide binding factor has not yet been identified *Stuve and Myers, 1990*. In the homozygous clones K and KDT, which harbor the edited KLF1 sequence, we observed increased binding of KLF1 to the *HBD* promoter relative to unedited WT HUDEP-2 cells. However, we found RNA Pol II binding at the *HBD* promoter binding only in the homozygous KDT clone. The homozygous DT clone, which harbors a TFIIB site, did not show significant binding of RNA Pol II. Taken together, our data show that three transcription factor binding sites are necessary to recruit the transcriptional machinery to the *HBD* promoter *Gillinder et al., 2018*.

We next performed qRT-PCR on all differentiated heterozygous and homozygous clones to look at percentage of *HBD* produced compared to total β-like globins (*Figure 2b*). *HBB* expression was unaffected by the K, DT, and KDT knock-ins (*Figure 2—figure supplement 1*). Like the pooled editing results, we found that clones with single knock-ins of KLF1 or β-DRF and TFIIB motifs alone did not show increased *HBD*. Strikingly, we observed a significant increase in *HBD* for KDT clones harboring all three motifs. Wildtype clones showed HBD expression less than 5% of β-like globins, while heterozygous KDT clones had increased *HBD* expressions ranging from 5% to 19% and homozygous KDT clones showed increases in *HBD* to ranges of 20–30% of β-like globins.

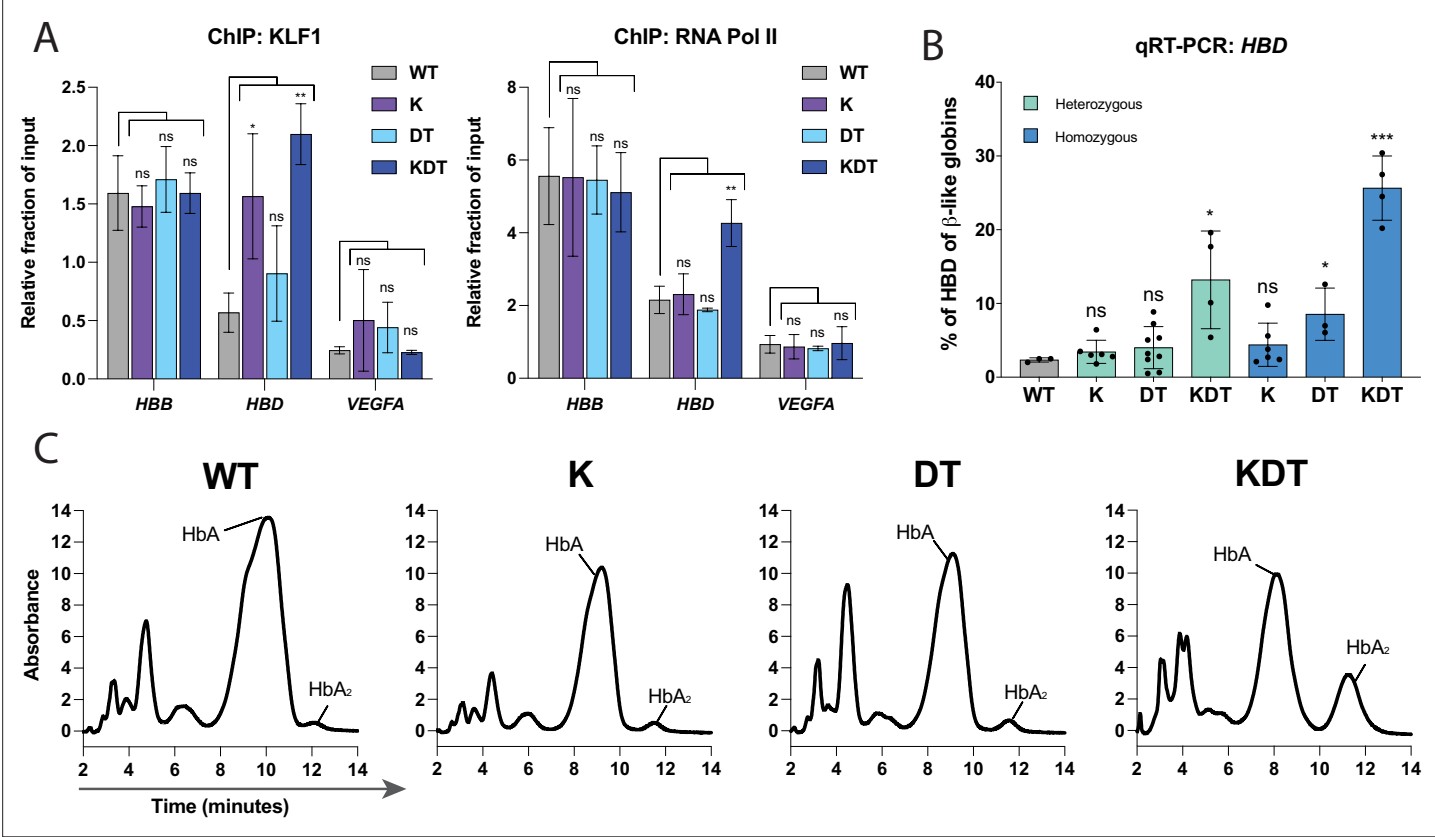

**Figure 2.** Characterization of HUDEP-2 clones with heterozygous or homozygous knock-ins of KLF1, β-DRF, and TFIIb sequences. (**A**) ChIP-qPCR of KLF1 and RNA Pol II performed on WT HUDEP-2 cells and homozygous clones of K, DT, and KDT. Data is shown as relative fraction of input and normalized to *SP1*. The genes targeted are *HBD*, *HBB*, and *VEGFA* as a negative control. Cells from WT, one K homozygous clone, one DT homozygous clone, and one KDT homozygous clone were grown and harvested separately and on different days for each biological replicate. The data is presented as mean ± SD of three biological replicates. P value indicates paired, two-tailed student *t* test (ns, non-significant; *, p≤0.05; **, p≤0.01). (**B**) qRT-PCR of *HBD* of HUDEP-2 heterozygous and homozygous clones with K, DT, and KDT knock-in and 5 days of differentiation. Each dot represents an individual clonal population, each validated by NGS. Data is plotted as % of β-like globins (*HBB, HBG1/2, HBD*). Each clone was differentiated and the data is presented as one replicate for each clonal population. p Value indicates paired, two-tailed student *t* test (ns, non-significant; *, p≤0.05; **, p≤0.01). (**C**) HPLC of of HUDEP-2 homozygous clones with K, DT, and KDT knock-in and 5 days of differentiation. Hemoglobin A (HbA) and Hemoglobin A2 (HbA2) peaks are annotated. HPLC of one homozygous clone of K, DT, and KDT was performed in triplicate, with a representative dataset of one replicate shown.

The online version of this article includes the following figure supplement(s) for figure 2:

**Figure supplement 1.** qRT-PCR of HUDEP-2 clones with homozygous knock-ins of KLF1, β-DRF, and TFIIb sequences.

Next, hemoglobin protein levels were measured using high pressure liquid chromatography (HPLC) for the differentiated homozygous clones (*Figure 2c*). Peaks were assigned to hemoglobin complexes based on previous work that performed mass-spectrometry on each globin peak fraction *Boontanrart et al., 2020*. We found that the transcript-level results are further supported at the protein level. The KLF1 and the β-DRF and TFIIB clones did not show any measurable increase in HbA2, while the KDT homozygous showed a large increase in HbA2.

## Endogenous editing of the *HBD* promoter increases *HBD* in CD34+ derived erythroblasts

To test the effects of *HBD* promoter engineering in a more clinically relevant cell type for a potential ex vivo therapy to treat β-hemoglobinopathies, we edited human CD34 +derived erythroblasts. We used CRISPR-Cas9 to perform pooled knock-in at the *HBD* promoter in mobilized peripheral blood CD34 +human hematopoietic stem and progenitor cells (HSPCs) with the K, DT, and KDT RNP and ssODNs. We found HDR rates ranging from 18% to 25% for all promoter knock-ins (*Figure 3a*). We differentiated the edited HSPC pools and performed qRT-PCR to measure *HBD* expression levels. As

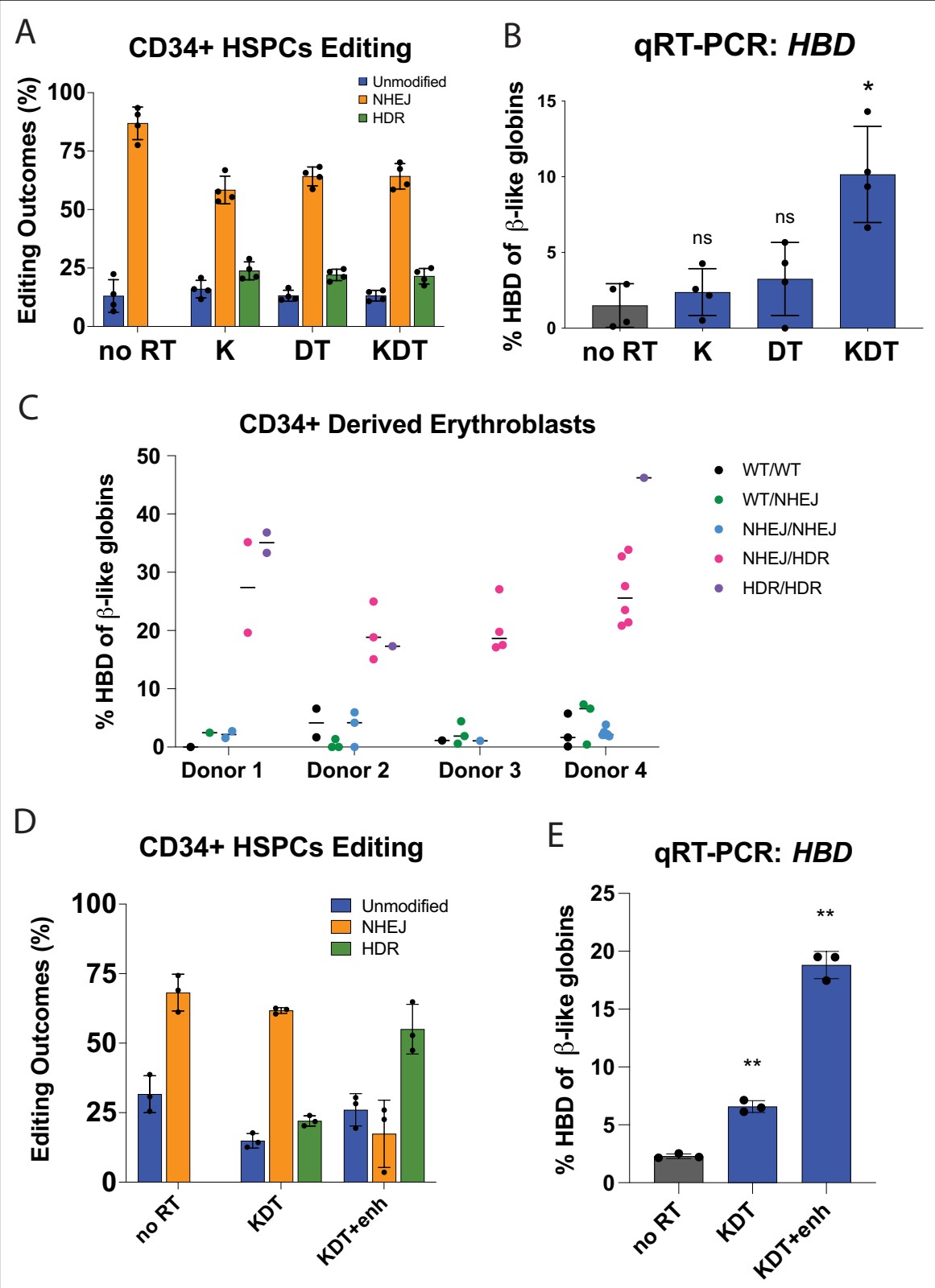

**Figure 3.** Endogenous editing of the *HBD* promoter in HSPCs. (**A**) Editing efficiencies showing percentages of unmodified, NHEJ, or HDR alleles. Conditions tested were Cas9 and sgRNA RNP with no repair template (no RT), K, DT, and KDT repair templates. The data is presented as independent editing experiments with four different donor samples. (**B**) qRT-PCR of *HBD* after pooled editing of HSPCs with no RT, K, DT, and KDT. Cells were expanded in erythroid expansion conditions and differentiated for 5 days. Data is plotted as % of all β-like globins (*HBB, HBG1/2, HBD*). The data is

*Figure 3 continued on next page*

*Figure 3 continued*

presented as independent editing experiments with four different donor samples. (**C**) qRT-PCR of *HBD* of clonal erythroblast populations after 5 days of differentiation. Genotypes were determined by NGS. Data is plotted as % of β-like globins (*HBB, HBG1/2, HBD*). The data is presented as independent editing experiments with four different donor samples and each dot denotes an individual clonal population. (**D**) Editing efficiencies showing percentages of unmodified, NHEJ, or HDR alleles. Conditions tested were Cas9 and sgRNA RNP with no repair template (no RT), KDT repair template, and KDT repair template with AZD-7648 (KDT +enh). The data is presented as one editing experiment with three different donor samples. (**E**) qRT-PCR of *HBD* after pooled editing of HSPCs with no RT, KDT, or KDT +enh. Cells were expanded in erythroid expansion conditions and differentiated for 5 days. Data is plotted as % of all β-like globins (*HBB, HBG1/2, HBD*). The data is presented as one editing experiment with three different donor samples.

The online version of this article includes the following figure supplement(s) for figure 3:

**Figure supplement 1.** Characterization of *HBD* promoter edited CD34 +HSPCs.

in HUDEP-2 cells, there was no statistically significant increase in *HBD* levels for K and DT, and a slight increase in four different donors in the KDT knock-in condition (*Figure 3b*).

Overall, we observed high editing rates in HSPCs using our CRISPR-Cas9 editing reagents. Cas9 editing can sometimes be accompanied by large deletions *Shin et al., 2017*, which in the case of editing the *HBD* promoter might impinge upon the neighboring gene for β-globin. To test if editing at the *HBD* promoter leads to decrease of other β-like globins, we edited four different HSPC donors with Cas9 and the *HBD* gRNA alone, mimicking a 'worst case' scenario of only NHEJ with no HDR alleles. We observed indel efficiencies ranging from 61% to 78% in the various donors (*Figure 3—figure supplement 1a*). We differentiated these edited pools into erythroblasts and performed qRT-PCR on the β-like globins *HBB, HBD*, and *HBG1/2* and normalized their expressions to *HBA* (*Figure 3—figure supplement 1b*). The *HBA* gene, which is present in a different locus from the β-like genes, should be unaffected by editing at the β-globin locus. We observed that *HBD* expression decreased by approximately 50%, while the expression of *HBB* and *HBG1/2* were not significantly changed. While we cannot rule out a small percentage of large deletions among the indel alleles, we do conclude that indel-based targeting of the *HBD* promoter decreases *HBD*, as expected, without grossly affecting the expression of the other β-like globins.

To test the effect of KDT editing on a single-cell level, we isolated and grew colonies from HSPCs under erythroid expansion conditions. Each colony was genotyped and their alleles classified as either unmodified, NHEJ, or HDR (*Figure 3—figure supplement 1c*). These CD34 +derived clonal erythroblasts were differentiated and qRT-PCR was performed to determine *HBD* expression relative to total β-like globins (*Figure 3D*). *HBB* transcript levels remain similar between unedited or NHEJ clones and KDT knock-in clones (*Figure 3—figure supplement 1d*). We observed that colonies with unedited or all NHEJ alleles had an average expression of lower than 3% *HBD* of total β-like globins. Heterozygous KDT knock-ins from multiple donors increased *HBD* expression dramatically, between 15–35% of total β-like globins in heterozygous knock-ins and between 17–46% of total β-like globins in homozygous knock-ins.

Clonal erythroblasts harboring either a heterozygous or homozygous knock-in of KDT expressed significantly increased *HBD*. However, the edited pools of HSPCs had only a slight increase in *HBD*, far below the level required to be clinically relevant. We hypothesized that increased HDR efficiency in the pool could lead to higher *HBD* levels. Small molecule drugs, most notably DNA-PKcs inhibitors that effectively inhibit NHEJ, have shown to shift editing outcomes in CRISPR-Cas9 induced double-stranded break to favor HDR *Lee et al., 2022*; *Peterka et al., 2022*. We tested the DNA-PKcs inhibitor AZD-7648 on HSPCs using our editing conditions for KDT knock-in. In three different HSPC donors, we observed an increase from 20% to 24% HDR alleles to 47–65% when we used AZD-7648. We differentiated the edited HSPC pools and performed qRT-PCR to measure *HBD* expression levels. In these edited erythroblast pools, the KDT-editing pool expressed 6–7% *HBD* of total β-like globins, while the pools edited with KDT along with HDR enhancer showed increased *HBD* expression to 17–19%, confirming that increasing the HDR efficiency of KDT knock-in can increase *HBD* levels within edited pools of HSPCs.

# Discussion

Current gene editing efforts to treat the β-hemoglobinopathies include correcting individual *HBB* mutations *DeWitt et al., 2016*, a method which would be limited to specific types of patient mutations, or increasing fetal hemoglobin *Wienert et al., 2018*, which has different oxygen-binding capacities than HbA1. In this study, we describe a path to upregulating HbA2, which shares high similarity to HbA1 and is applicable to all β-hemoglobinopathy disease mutations.

Previous transgenic studies have shown that insertion of KLF1 alone to the *HBD* promoter sequence is sufficient to drive expression of *HBD Porcu et al., 2021*; *Donze et al., 1996*. In our study, we found that endogenous insertion of a KLF1 motif is insufficient to drive *HBD* expression. One explanation of this discrepancy could be that previous transgenic studies do not reflect the complex chromatin context and regulation at the β-like globin locus. Another explanation could be the importance of the KLF1 motif placement. To ablate re-recognition of the HDR alleles by Cas9, we inserted the KLF1 motif in the K repair template after the TATA box, while the DT and KDT repair templates inserted all motifs upstream of the TATA box. In the *HBB* promoter, the KLF1 motif is upstream of the TATA box. In a previous study using a HBD promoter luciferase reporter system, they found the highest luciferase activity when the KLF1 motif was upstream of the TATA box, in the homologous position to the HBB promoter *Donze et al., 1996*. In another study, it was shown that insertion of the KLF1 motif in varying positions resulted in differences in luciferase activity *Ristaldi et al., 1999*. The placement of transcription factor binding sites can play a crucial role in promoter activity and follow-up studies of our work should further test the spacing and positioning of these various promoter elements, as well as include the NF-Y motif that was not explored in this study.

We have shown that insertion of three motifs, KLF1, β-DRF, and TFIIB, is necessary to recruit RNA Pol II and induce high expression of *HBD*. In CD34 + derived erythroblasts, homozygous knock-in of KLF1, β-DRF, and TFIIB leads to increases in *HBD* up to 46% of total β-like globins in clonal populations. To our knowledge, this is the first genomic editing of the *HBD* promoter that results in increased HbA2. Bulk editing of *HBD* through insertion of multiple transcriptional elements will rely on high levels of HDR. Increasing HDR outcomes might be achieved in CD34 + cells by a variety of methods such as controlled cell-cycling or modulation of DNA repair factors *Howden et al., 2016*; *Charpentier et al., 2018*; *Schiroli et al., 2019*; *Shin et al., 2018*. In this study, we found that AZD-7648 increased HDR outcomes in pooled edited HSPCs with concomitant increases in *HBD* expression. Further work should assess the safety and the effects of the AZD-7648 or other HDR-enhancing in HSPCs if they are to be pursued in a clinical context.

We observed high editing efficiencies when targeting CD34 + HSPCs. When editing targets *HBB*, NHEJ alleles that co-occur with HDR alleles can cause reductions in overall β-globin levels *DeWitt et al., 2016*. When targeting *HBD*, *HBB* might also be inadvertently affected by large deletions that extend into neighboring β-like globin genes. However, we using qRT-PCR experiment we found that *HBB* and *HBG1/2* mRNA expression were unaffected in our edited populations of HSPCs, suggesting that large deletions were not occurring with enough frequency to grossly affect β-like globin expression. However, some studies have shown that large deletions can occur at low levels, which might be detected by very sensitive tests such as long-range next-generation sequencing or ddPCR.

It is estimated that roughly 30% of beneficial hemoglobin (10 pg per cell) is sufficient to ameliorate β-hemoglobinopathy symptoms *Steinberg, 2021*. There are currently many gene editing approaches being explored for the β-hemoglobinopathies, mainly focused on increasing fetal hemoglobin or directly correcting the SCD mutation *Wienert et al., 2018*. A recent study reversed the SCD mutation using CRISPR-Cas9 prime-editing *Everette et al., 2023*. They showed SCD mutation reversion rates of 17–41% in 4 SCD patient donors and resulted in expression of HbA1 to 10–45% of total hemoglobins. Sickling decreased proportionally to editing efficiencies, with an average of 63% sickled cells for untreated samples and decreasing to 37% in treated samples. Another strategy utilizes CRISPR-Cas9 base-editing to disrupt the LRF repressor binding site within *HBG1/2* and observed an upregulation of HbF to over 20% of total hemoglobins in SCD HSPCs *Antoniou et al., 2022*. When disrupting the LRF site by insertion of a KLF1 site, HbF levels increased to 65–77%. All edited cell populations showed a therapeutically relevant reduction in the number of sickling cells. Another approach utilizes CRISPR-Cas9 to disrupt the enhancer region of BCL11A, a repressor of fetal hemoglobin *Frangoul et al., 2021*. In an on-going clinical trial, BCL11A enhancer targeting yields editing efficiencies of 80–93% in four SCD patients and HbF upregulation to over 30% of total hemoglobins, with no

vaso-occlusive crises 3 months post-transfusion of edited cells. These studies support the curative effect of increasing functional hemoglobin levels on β-hemoglobinopathies. In our study using CRISPR-Cas9 and a DNAPKcs inhibitor to knock the KDT promoter into *HBD*, we observed increased HbA2 to 17–19% of total hemoglobins in CD34+-derived erythroblasts. Further studies will need to be done to achieve even higher levels of HbA2 expression or test whether this lower expression of HbA2 is sufficient to ameliorate a disease phenotype.

In our experiments, heterozygous and homozygous knock-in of KDT in CD34+ erythroblasts led to 15–46% *HBD* relative to total β-like globins. Interestingly, in edited CD34+ erythroblasts, we observed that heterozygous and homozygous KDT populations expressed similar increases in *HBD*. Further studies should investigate whether heterozygous knock-in of KDT in β-hemoglobinopathy cells is sufficient to ameliorate disease phenotypes. For example, one could knock-in KDT to the *HBD* promoter of SCD patient HSPCs and perform HPLC or microscopy assays to determine the anti-sickling effects of a heterozygous or homozygous knock-in. If heterozygous knock-in of KDT is sufficient, one would see a decrease in sickle hemoglobin produced, or a decreased percentage of sickled red blood cells.

In summary, our presented study provides a novel strategy to increase levels of HbA2 from the endogenous *HBD* locus that could potentially be applicable as an ex vivo gene therapy. In pre-clinical studies, it will be important to quantitatively investigate the amount of *HBD* that is optimal for improving the function and health of patient red blood cells and explore the overall potential and safety of increasing HbA2 as a therapeutic option for the β-hemoglobinopathies.

# Materials and methods

## Cas9 RNP nucleofection
Cas9 RNP was performed as described previously (*Lingeman et al., 2017*). Briefly, either IVT guides are purified or chemically protected guides were ordered from Synthego and complexed with purified Cas9-NLS protein. The nucleofection was performed using Lonza 4D-Nucleofector and using the P3 Primary Cell 96-well NucleofectorTM Kit (V4SP-3096) following manufacturer's instructions. The HUDEP-2 nucleofector code used was DD-100 and for primary HSPCs ER-100.

## IVT sgRNA
Guide RNAs (*Supplementary file 1*) were in vitro transcribed as described previously (*Lingeman et al., 2017*). Briefly, guide sequences were ordered as oligonucleotides and formed into duplexes using a PCR thermocycler. The DNA template was transcribed to RNA using HiScribe T7 High Yield RNA Synthesis Kit (E2040S) following manufacturer protocol. The resulting RNA was purified using RNeasy Mini kit (74104) and Rnase-Free DnaseI Kit (79254).

## High-pressure liquid chromatography and mass Spectrometry
HUDEP-2 cells or HSPCs were differentiated and harvested for lysis in hemolysate reagent containing 0.005 M EDTA and 0.07% KCN at 10,000 cells per microliter. The lysis was incubated at room temperature for 10 min and then centrifuged at max speed for 5 min. The supernatant was collected and run on Agilent 1260 Infinity II using a PolyCAT A column from PolyLC, 35x4.6 mm (3 µm;1500 Å) Serial# B19916E. The following Buffer compositions were used: Mobile Phase A: 20 mM Bis-tris, 2 mM NaCN pH 6.8 and Mobile Phase B: 20 mM Bis-tris, 2 mM NaCN, 200 mM NaCl, pH 6.9. The following flow settings were used: Gradient: 0–8' 2–25% Phase B, 8–18' 25–100% Phase B, 18–23' 100–2% Mobile Phase B using a Flow Rate: 1.5 mL/min and measuring detection of 415 nm Diode Array.

## HUDEP-2 cell culture and differentiation
All cell culture was performed at 37 °C in a humidified atmosphere containing 5% CO2. HUDEP-2 cells (RRID: CVCL_VI06), obtained from the Riken Institute *Kurita et al., 2013*, were tested negative for mycoplasm and the cell line was authenticated by STR profiling. HUDEP-2 cells were cultured in a base medium of SFEM (Stemcell Technologies 9650) containing to a final concentration of dexamethasone 1 µM (Sigma D4902-100MG), doxycycline 1 µg/ml (Sigma D9891-1G), human stem cell factor 50 ng/ml (PeproTech 300–07), erythropoietin 50 ng/ml (Peprotech 100–64), and penstrept 1%. Cells were cultured at a density of 2e5 – 1e6 cells/ml. For differentiation, HUDEP-2 cells are centrifuged at 500 *g* for 5 min, media is removed and replaced with differentiation media. Differentiation

media consists of a base media of IMDM + Glutamax (ThermoFisher 31980030) containing to a final concentration human serum 5% (Sigma H4522-100mL), heparin 2 IU/ml (Sigma H3149-25KU), insulin 10 µg/ml (Sigma I2643-25mg), erythropoietin 50 ng/ml (Peprotech 100–64), holo-transferrin 500 µg/ml (Sigma T0665-100mg), mifepristone 1 µM (Sigma M8046-100MG), and doxycyline 1 µg/ml (Sigma D9891-1G). Cells are differentiated for 5 days and then harvested for analysis.

## mPB-HSPCs cell culture and differentiation

For editing for human CD34 + cells, CD34 + mobilized peripheral blood HSPCs were thawed and cultured in SFEM containing CC110 supplement (Stemcell Technologies 02697) for 2 days. CD34 + cells were then electroporated and transferred into erythroid expansion media containing SFEM and erythroid expansion supplement (Stemcell Technologies 02692) for 7 days and cultured at a density of 2e5-1e6 cells/ml. The resulting early erythroblasts were transferred to differentiation media containing SFEM with 50 ng/ml erythropoietin, 3% normal human serum, and 1 µM mifepristone. The resulting late erythroblasts were harvested for analysis after 5 days of differentiation. For generation of clonal erythroblasts, CD34 + cells were then electroporated and transferred into erythroid expansion media containing SFEM and erythroid expansion supplement (Stemcell Technologies 02692) for 4 days. The early erythroblasts were then single cell seeded and cultured for 7 days. After 7 days, the cells are transferred into differentiation media containing SFEM with 50 ng/ml erythropoietin, 3% normal human serum, and 1 µM mifepristone. The resulting late erythroblasts were harvested for analysis after 5 days of differentiation.

## qRT-PCR

RNA was harvested from cells using Qiagen RNeasy Mini Kit and Rnase-Free DnaseI Kit following manufacturer's instructions. RNA was reverse transcribed to cDNA using Iscript Reverse Transcription Supermix (Bio-Rad) and qRT-PCR reactions were set up using SsoAdvanced Universal SYBR Green or SsoFast EvaGreen Supermix (Bio-Rad). Reactions were run on the StepOne Plus Real-Time PCR System (Applied Biosystems) or the QuantStudio 6 Flex (Thermo Fisher). Samples were analyzed using a two-step amplification and melt curves were obtained after 40 cycles. The Ct values for genes of interest were normalized to GAPDH, and expressions of genes are represented as 2-[ΔCt] or 2-[ΔΔCt] for fold change over control condition. All primers used for qRT-PCR are listed in *Supplementary file 1*.

## ChIP-qPCR

ChIP was performed as done previously *Boontanrart et al., 2020*. Briefly, 10 million cells per sample were harvested and cross-linked in 1% Formaldehyde. Cross-linking was quenched with the addition of 1.5 M glycine. Samples were then lysed for 10 min at 4 °C in 50 mM Hepes–KOH, pH 7.5; 140 mM NaCl; 1 mM EDTA; 10% glycerol; 0.5% NP-40 or Igepal CA-630; 0.25% Triton X-100. Cells were then centrifuged at 1500 $g$ for 3 min and the supernatant was discarded. The pellet was resuspended in 10 mM Tris–HCl, pH8.0; 200 mM NaCl; 1 mM EDTA; 0.5 mM EGTA and incubated for 5 min at 4 °C. The cells were then centrifuged at 1500 g for 3 min and the supernatant was discarded. The pellet was resuspended in 10 mM Tris–HCl, pH 8; 100 mM NaCl; 1 mM EDTA; 0.5 mM EGTA; 0.1% Na–Deoxycholate; 0.5% N-lauroylsarcosine and sonicated using the Covaris S220 following manufacturer's instructions. Protein G beads (ThermoFisher) were complexed with antibody and the antibody-bead complexes were incubated with cell lysates at 4 C overnight with rotation. The antibodies used were mouse anti-RNA Pol II (Diagenode C15100055-100 RRID: AB_2750842) and goat anti-KLF1 (Origene TA305808). The beads were retrieved using a magnetic stand and rinsed with RIPA buffer. Elution buffer containing 50 mM Tris–HCl, pH 8; 10 mM EDTA; 1% SDS was added to the beads for reverse crosslinking at 65 °C overnight with shaking. After reverse crosslinking, the beads were removed. The eluted DNA was treated with RNaseA and Proteinase K and then purified using Qiagen MinElute PCR Purification Kit, following the manufacturer's instructions. qPCR reactions were set up using SsoAdvanced Universal SYBR Green or SsoFast EvaGreen Supermix (BioRad). Reactions were run on the StepOne Plus Real-Time PCR System (Applied Biosystems) or the QuantStudio 6 Flex (Thermo Fisher). The Ct values were analyzed by the relative fraction of input method.

## Next-Generation Sequencing (NGS) amplicon preparation and analysis

To confirm homozygosity/heterozygosity of the *HBD* clones, samples' gDNA extracted with QuickExtract DNA Kit was first amplified by PCR. The primers were designed specifically for NGS, spanning a region of <200 bp (including the primers sequences) in which the cutsite is asymmetrically placed (e.g., 30–80 bp from the forward or the reverse primer) to capture the edited region. Subsequently, two stubber sequences are added, one for the forward primer (5'– CTTTCCCTACACGACGCTCT TCCGATCT –3') and one for the reverse (5'– GGAGTTCAGACGTGTGCTCTTCCGATCT –3'). After running the first PCR to amplify the genetic region of interest, the overhanging stubber sequences are used to run a second PCR with indexing primers (forward and reverse primers are premixed at 5 M). Samples are pooled and purified with SPRIselect beads, 5 mL (Beckman Coulter, B23317) using a DynaMag–2 Magnet magnetic stand (Thermo Fisher Scientific, 12321D). NGS is performed by the Genome Engineering and Measurement Lab (GEML, ETH Zürich) using a NovaSeq 6000 Sequencing System (Illumina, 20012850) or MiSeq. Sequencing mode used was either 100 or 150 PE (paired end). Editing efficiencies were determined using CRISPResso *Pinello et al., 2016*.

## Acknowledgements

We thank laboratory members for helpful discussions and support; D Fercher and M Zenobi-Wong for help with HPLC; R Kurita and Y Nakamura for their contribution of HUDEP-2 cells; Genome Engineering Measurement Lab and Functional Genomics Center Zurich (ETH Zurich/University of Zurich) for help in running NGS samples. We also acknowledge the Cooperative Centers of Excellence in Hematology NIDDK Grant # DK106829 for CD34 +HSPCs. JEC is supported by the NOMIS Foundation and the Lotte and Adolf Hotz-Sprenger Stiftung. MYB is supported by ETH Foundation's ETH Pioneer Fellowship and the SNSF BRIDGE Foundation.

## Additional information

### Funding

| Funder | Grant reference number | Author |
| --- | --- | --- |
| ETH Zürich Foundation | ETH Pioneer Fellowship | Mandy Y Boontanrart |
| SNSF | BRIDGE Proof of Concept | Mandy Y Boontanrart |

The funders had no role in study design, data collection and interpretation, or the decision to submit the work for publication.

### Author contributions

Mandy Y Boontanrart, Conceptualization, Formal analysis, Supervision, Investigation, Visualization, Writing – original draft; Elia Mächler, Simone Ponta, Jan C Nelis, Viviana G Preiano, Investigation; Jacob E Corn, Supervision, Writing – review and editing

### Author ORCIDs

Simone Ponta http://orcid.org/0000-0002-0007-0346
Jacob E Corn http://orcid.org/0000-0002-7798-5309

### Decision letter and Author response

Decision letter https://doi.org/10.7554/eLife.85258.sa1
Author response https://doi.org/10.7554/eLife.85258.sa2

## Additional files

### Supplementary files

• Supplementary file 1. Oligo sequences used for the HBD IVT templates for the generation of sgRNAs, repair template sequences, NGS primers, qRT-PCR primers, and ChIP qRT-PCR primers.

• MDAR checklist

## Data availability

All data generated are included in the manuscript. All DNA sequences and oligo information are listed in Supplemental Table 1.

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
