## [Editor Report]

This study presents the important finding that gene editing could be used to activate δ-globin expression to treat disorders of red blood cell synthesis. The evidence supporting the claims of the authors is convincing, particularly in the clonal cell lines. The data show this approach to have promise and identify avenues of effort that could be pursued to advance it to a clinical strategy for hemoglobinopathy treatment.

---

## [Decision Letter]

**Decision letter after peer review:**

Thank you for submitting your article "Engineering of the Endogenous HBD promoter increases HbA2" for consideration by *eLife*. Your article has been reviewed by 3 peer reviewers, one of whom is a member of our Board of Reviewing Editors, and the evaluation has been overseen by Didier Stainier as the Senior Editor. The following individual involved in the review of your submission has agreed to reveal their identity: Fyodor D. Urnov (Reviewer #2).

Essential revisions:

1) The data presented in Figure 3C appears to be derived from a small number of clonal populations of cells from two donors. For some genotypes, data from a single colony only is presented. A larger dataset for this experiment would greatly improve the strength of conclusions that are able to be drawn from this data (Reviewer 2).

2) The strength of the conclusions in the manuscript should be altered to reflect the observation that editing of all three promoter elements in populations of cells is far below the level required to be clinically relevant and to suggest ways that this could be improved (Reviewer 3; Reviewer 2; Reviewer 1, Comment 3)

3) The high NHEJ seen in the CD34+ cells would be likely to lead to deletions that could inactivate genes of the β-globin locus and cause reductions in the total levels of β-like globins resulting in thalassemic cells. This caveat of the study and the implications of this high NHEJ for the therapeutic translation of this approach should be discussed in detail (Reviewer 3)

4) A section should be added to the discussion to comment on the level of HbA2 seen via the strategy in this manuscript and how it compares to other genome editing approaches currently being explored for the β-hemoglobinopathies (Reviewer 2)

5) It should be made clear throughout the manuscript that the KLF1 site has been inserted at a different position to the position of this site in the HBB promoter and in the K v KDL edited cells (Reviewer 1, Comment 1)

6) The justification for the exclusion of the CP1/NF-Y site from this study should be expanded (Reviewer 1, Comment 2)

*Reviewer #1 (Recommendations for the authors):*

This is a thorough and interesting paper. I have provided a positive overall review of the manuscript in the public review. Here, I will not repeat those comments but instead, provide specific suggestions on how the manuscript could be strengthened. Please note that addressing these comments would not require additional experimental work to be performed, so these are relatively minor concerns and do not detract from my enthusiasm for the work.

Comments:

1. In reference to the finding (quoted from the abstract) "We find that insertion of a KLF1 site alone is insufficient to upregulate δ-globin.": it is important to note that the KLF1 site has been inserted at a different position in the K edited cells (downstream of the TATA box) to the position in the KDL edited cells (just upstream of the β-DRF box) and both of these are different to that of the promoter location of the KLF1 site in the β-globin promoter (upstream of the NF-Y site). The spacing and positioning of promoter elements are important, and the placement chosen for insertion of the KLF1 sites in this study should be presented with a clear rationale so that readers of this manuscript can better appreciate this:

a. It should be made clear in the abstract that the KLF1 sites have been inserted at a de novo/non-homologous position (when compared to the location of the site in the HBB promoter).

b. It should be made explicitly clear in the Results section that the location of the KLF1 sites is different from that in the HBB promoter and different in the K v KDT cells and a justification for this choice provided.

c. The sentence "We maintained the ordering between motifs to mimic the HBB promoter and paid particular attention to limiting the number of mutated base pairs to increase the likelihood of successful HDR editing." should be amended. While the order of the K, D, and T elements relative to each other is maintained in the KDT-edited cells, the K element is in two different positions, neither of which is in the homologous position to the KLF1 site in the HBB promoter.

d. The statement "Taken together, these data indicate that an intact KLF1 site is sufficient to recruit KLF1…" should be modified. The experiments performed do not repair the KLF1 site to create an intact KLF1 site. Instead, they introduce a KLF1 site at a new location (or, two new locations).

e. A section should be added to the discussion where the data from this study is compared and contrasted to that from references 6, 16, and 17, where, as the authors state "Previous studies using transgenic approaches have shown that inclusion of a KLF1 motif in the HBD promoter can drive exogenous expression of δ-globin". The different locations of the KLF1 site in the K v KDT cells and how this could influence results should also be commented on. In my opinion, the way the KLF1 data is presented in this paper could result in the importance of the KLF1 site alone being inappropriately dismissed if this greater context is not provided.

2. In reference to the CP1 site:

a. It is my understanding that the promoter site referred to as the CP1 site (in the Results section) is more commonly referred to in the field as an NF-Y site. This should be amended.

b. The statement "A previous transgenic study showed that the CP-1 site plays a minor role in HBB transcription, and was therefore omitted from our designs21" is misleading. While, naturally, it would not have been practicable to target every promoter element in every combination in this study, I feel that the potential importance of this site has been inappropriately dismissed. In reference 21 (doi: 10.1182/blood.V90.1.421) the authors saw upregulation of δ-globin promoter activity in K562 cells and MEL cells when the CCAAT box was repaired and saw a reduction in β-globin expression when the CCAAT box was mutated (Figure 3). This is not consistent with the statement that the authors of this paper have made while citing this work. While this old study has its caveats, including that these experiments were luciferase reporter assays. A different rationale for why the NF-Y site was not considered should be presented.

c. In the Discussion section (perhaps in a new section where KLF1 site placement could be discussed, see above) the potential inclusion of other promoter elements/edits, such as the NF-Y site, in future studies to yield more robust potentially-therapeutic δ-globin upregulation could be discussed.

3. In reference to the abstract statement "Edited CD34+ hematopoietic stem and progenitors (HSPCs) differentiated to primary human erythroblasts express up to 35% HBD." the findings should be presented slightly more conservatively:

a. It should be made clear in the abstract that this is in clonal populations. The effect is not seen in pooled populations.

b. It should be made clear in the introduction that this is in clonal populations. The effect is not seen in pooled populations. For example, the sentence "However, insertion of KLF1, β-DRF, and TFIIB motifs drive high expression of δ-globin from the endogenous locus in HUDEP-2 cells and primary erythroblasts" should be modified.

c. It should be made clear in the discussion that this is in clonal populations of edited cells. For example, in the sentence "Our promoter engineering approach dramatically increases HBD levels…" and the sentence "In our experiments, heterozygous and homozygous knock-in of KDT in CD34+ erythroblasts led to 15 – 37% HBD relative to total β-like globins."

d. It should be made clear in the Discussion section that while this data is mechanistically interesting and encouraging, further work will be necessary to yield either higher % HDR or higher HBD upregulation levels (in edited cells) for this to be therapeutically useful in pooled populations of patient cells (and thus therapeutically useful). HBD upregulation is an important first step but the therapeutic benefit of HBD upregulation has not been directly addressed in this manuscript.

Alternatively, additional experiments with cells from more donors could be performed to allow these original statements to be better supported.

---

## [Author Response]

Essential revisions:1) The data presented in Figure 3C appears to be derived from a small number of clonal populations of cells from two donors. For some genotypes, data from a single colony only is presented. A larger dataset for this experiment would greatly improve the strength of conclusions that are able to be drawn from this data (Reviewer 2).

We have now performed more experiments to strengthen Figure 3. As the reviewers suggested, we have isolated more edited HSPC colonies for Figure 3C. Specifically, we edited two more donors and isolated single colonies from here. As before, per donor, some genotypes only have few colonies represented. This is due to the difficulty in growing single cell CD34+ derived colonies large enough for genotyping and RNA extraction.

Additionally, we have also increased Figure 3A and 3B from n=2 HSPC donors to n=4 HSPC donors. Similarly, Figure 2B originally had one heterozygous or homozygous clone per knock-in condition, and we have now isolated more clones, at least three per condition.

2) The strength of the conclusions in the manuscript should be altered to reflect the observation that editing of all three promoter elements in populations of cells is far below the level required to be clinically relevant and to suggest ways that this could be improved (Reviewer 3; Reviewer 2; Reviewer 1, Comment 3)

In the Results section on HSPCs, we have now edited the text to state that KDT knock-in at the pooled level yields only slightly increases in HBD and that is below the level to be clinically relevant. We also state that in order to improve HBD levels, the HDR efficiency of knocking in KDT must be increased. We proposed using small molecule drugs and tested one HDR enhancer drug AZD4678. We find that AZD46789 is able to increase HDR efficiency with concomitantly increased HBD expression. This data is now included as new figures 3D and 3E. We have also edited the text in the Discussion section to state that further studies on the effects of AZD4678 will be necessary before utilizing it in a clinical context.

3) The high NHEJ seen in the CD34+ cells would be likely to lead to deletions that could inactivate genes of the β-globin locus and cause reductions in the total levels of β-like globins resulting in thalassemic cells. This caveat of the study and the implications of this high NHEJ for the therapeutic translation of this approach should be discussed in detail (Reviewer 3)

We were not completely sure about the origin of this point, since our edits are aimed at HBD, which makes up less than 5% of total hemoglobins under normal conditions. NHEJ occurring in HBB (e.g. when doing HDR for direct correction) would potentially yield thalassemic cells. But indels in the HBD promoter might at most cause a 5% decrease in total globin levels (if δ expression was completely destroyed). We have performed a new experiment to explicitly address this point. We edited n=4 CD34+ HSPCs donors and compared unedited populations to populations edited with Cas9+HBD gRNA but no repair template. This represents a “worst case” scenario, in which there can be no HDR-based promoter engineering and only NHEJ. These data are included this in Supplementary Figure 3. We observed high editing efficiency of 61 – 78% in the HBD promoter. We performed qRT-PCR of the β-like globins in edited pools and normalized to HBA, reasoning that HBA is a neutral control for absolute levels of each globin in the β locus because HBA is located in a different locus. By qRT-PCR, HBD transcripts were decreased by half compared to mock treated cells, while HBB and HBG1/2 were non-significantly affected. But as mentioned above, HBD expression makes up less than 5% of total hemoglobins, and therefore a half reduction in HBD represents a total reduction of 2.5% of globins. We do acknowledge that this experiment does not specifically quantify the rates of large deletions that might span from δ to β, and further studies would be needed to address this point. But if such large deletions do exist, they do not greatly affect β expression. We have included this in the results and the Discussion section.

4) A section should be added to the discussion to comment on the level of HbA2 seen via the strategy in this manuscript and how it compares to other genome editing approaches currently being explored for the β-hemoglobinopathies (Reviewer 2)

We have now added a new section in the discussion summarizing some of the recent genome editing approaches for hemoglobinopathies. Specifically, we mention CRISPR Therapeutics’ clinical trial on the BCL11A enhancer, David Liu’s most recent paper on base-editing to correct the SCD mutation, and Annarita Miccio’s recent paper on disrupting a repressor binding site on the γ-globin promoter.

5) It should be made clear throughout the manuscript that the KLF1 site has been inserted at a different position to the position of this site in the HBB promoter and in the K v KDL edited cells (Reviewer 1, Comment 1)

We have now edited the Results section and the Discussion section to clarify this important point.

6) The justification for the exclusion of the CP1/NF-Y site from this study should be expanded (Reviewer 1, Comment 2)

We have expanded this justification in the Results section as well as included this in the Discussion section as an important future study.

Reviewer #1 (Recommendations for the authors):This is a thorough and interesting paper. I have provided a positive overall review of the manuscript in the public review. Here, I will not repeat those comments but instead, provide specific suggestions on how the manuscript could be strengthened. Please note that addressing these comments would not require additional experimental work to be performed, so these are relatively minor concerns and do not detract from my enthusiasm for the work.

We would like to thank reviewer 1 for their very comprehensive insight in to our manuscript. Their comments and suggestions were insightful and helped us to think of important follow-up studies for future exploration. Reviewer 1 has a deep interest in the hemoglobin genetics field and we hope that they find their comments below well-addressed.

Comments:1. In reference to the finding (quoted from the abstract) "We find that insertion of a KLF1 site alone is insufficient to upregulate δ-globin.": it is important to note that the KLF1 site has been inserted at a different position in the K edited cells (downstream of the TATA box) to the position in the KDL edited cells (just upstream of the β-DRF box) and both of these are different to that of the promoter location of the KLF1 site in the β-globin promoter (upstream of the NF-Y site). The spacing and positioning of promoter elements are important, and the placement chosen for insertion of the KLF1 sites in this study should be presented with a clear rationale so that readers of this manuscript can better appreciate this:a. It should be made clear in the abstract that the KLF1 sites have been inserted at a de novo/non-homologous position (when compared to the location of the site in the HBB promoter).b. It should be made explicitly clear in the Results section that the location of the KLF1 sites is different from that in the HBB promoter and different in the K v KDT cells and a justification for this choice provided.

This is an important point to make and we have now addressed these points a. and b. in the results and the Discussion section.

c. The sentence "We maintained the ordering between motifs to mimic the HBB promoter and paid particular attention to limiting the number of mutated base pairs to increase the likelihood of successful HDR editing." should be amended. While the order of the K, D, and T elements relative to each other is maintained in the KDT-edited cells, the K element is in two different positions, neither of which is in the homologous position to the KLF1 site in the HBB promoter.

We have added to this paragraph considerably to address the motif spacing and placements of the repair template designs.

d. The statement "Taken together, these data indicate that an intact KLF1 site is sufficient to recruit KLF1…" should be modified. The experiments performed do not repair the KLF1 site to create an intact KLF1 site. Instead, they introduce a KLF1 site at a new location (or, two new locations).

We have now modified this statement in the text.

e. A section should be added to the discussion where the data from this study is compared and contrasted to that from references 6, 16, and 17, where, as the authors state "Previous studies using transgenic approaches have shown that inclusion of a KLF1 motif in the HBD promoter can drive exogenous expression of δ-globin". The different locations of the KLF1 site in the K v KDT cells and how this could influence results should also be commented on. In my opinion, the way the KLF1 data is presented in this paper could result in the importance of the KLF1 site alone being inappropriately dismissed if this greater context is not provided.

We appreciate bringing this important point to our attention and have now added this to the discussion.

2. In reference to the CP1 site:a. It is my understanding that the promoter site referred to as the CP1 site (in the Results section) is more commonly referred to in the field as an NF-Y site. This should be amended.

We have now changed all references to CP1 in the figures and text to NF-Y.

b. The statement "A previous transgenic study showed that the CP-1 site plays a minor role in HBB transcription, and was therefore omitted from our designs21" is misleading. While, naturally, it would not have been practicable to target every promoter element in every combination in this study, I feel that the potential importance of this site has been inappropriately dismissed. In reference 21 (doi: 10.1182/blood.V90.1.421) the authors saw upregulation of δ-globin promoter activity in K562 cells and MEL cells when the CCAAT box was repaired and saw a reduction in β-globin expression when the CCAAT box was mutated (Figure 3). This is not consistent with the statement that the authors of this paper have made while citing this work. While this old study has its caveats, including that these experiments were luciferase reporter assays. A different rationale for why the NF-Y site was not considered should be presented.

We have now modified the justification statement for omitting NF-Y and have also included it in the Discussion section on further testing of placements and spacing of the motifs in the repair templates. The sentence we use now is: “A previous transgenic study showed that the NF-Y site plays a lesser role in *HBB* transcription in comparison to KLF1^21^, and in the interest of reducing how many transcription element iterations to test, we therefore omitted NF-Y from our designs.”

We maintain that NF-Y plays a lesser role compared to KLF1 based on the reference 21 where they showed and stated in the abstract that restoring the NF-Y site led to increased luciferase activity, but restoration of KLF1 site led to much higher luciferase activity:

“Furthermore, a plasmid containing a single base pair (bp) mutation in the CCAAC box of the δ promoter, restoring the CCAAT box, caused a 5.6-fold and 2.4-fold (P <.05) increase of LUC activity in transfected K562 cells and MEL cells, respectively, in comparison to the wild-type δ promoter. A set of substitutions that create an EKLF binding site centered at -85 bp increased the expression by 26.8-fold and 6.5-fold (P <.05) in K562 and MEL cells, respectively.”

c. In the Discussion section (perhaps in a new section where KLF1 site placement could be discussed, see above) the potential inclusion of other promoter elements/edits, such as the NF-Y site, in future studies to yield more robust potentially-therapeutic δ-globin upregulation could be discussed.3. In reference to the abstract statement "Edited CD34+ hematopoietic stem and progenitors (HSPCs) differentiated to primary human erythroblasts express up to 35% HBD." the findings should be presented slightly more conservatively:a. It should be made clear in the abstract that this is in clonal populations. The effect is not seen in pooled populations.b. It should be made clear in the introduction that this is in clonal populations. The effect is not seen in pooled populations. For example, the sentence "However, insertion of KLF1, β-DRF, and TFIIB motifs drive high expression of δ-globin from the endogenous locus in HUDEP-2 cells and primary erythroblasts" should be modified.c. It should be made clear in the discussion that this is in clonal populations of edited cells. For example, in the sentence "Our promoter engineering approach dramatically increases HBD levels…" and the sentence "In our experiments, heterozygous and homozygous knock-in of KDT in CD34+ erythroblasts led to 15 – 37% HBD relative to total β-like globins."d. It should be made clear in the Discussion section that while this data is mechanistically interesting and encouraging, further work will be necessary to yield either higher % HDR or higher HBD upregulation levels (in edited cells) for this to be therapeutically useful in pooled populations of patient cells (and thus therapeutically useful). HBD upregulation is an important first step but the therapeutic benefit of HBD upregulation has not been directly addressed in this manuscript.Alternatively, additional experiments with cells from more donors could be performed to allow these original statements to be better supported.

We have addressed points a-d above and modified the statements in the manuscript. We have added a section in the results and discussion to address further work needed to increase HDR % and performed an experiment to demonstrate.